# Age-Dependent Changes in the Occurrence and Segregation of GABA and Acetylcholine in the Rat Superior Cervical Ganglia

**DOI:** 10.3390/ijms25052588

**Published:** 2024-02-23

**Authors:** Alfredo Hernández, Constanza González-Sierra, María Elena Zetina, Fredy Cifuentes, Miguel Angel Morales

**Affiliations:** Departamento de Biología Celular and Fisiología, Instituto de Investigaciones Biomédicas, Universidad Nacional Autónoma de México, Ciudad de México 04510, Mexico; alfredoherg@gmail.com (A.H.); cmarygonsierra@gmail.com (C.G.-S.); malez@iibiomedicas.unam.mx (M.E.Z.); fcifuent@iibiomedicas.unam.mx (F.C.)

**Keywords:** development, neurotransmitters, cotransmission, aging, sympathetic

## Abstract

The occurrence, inhibitory modulation, and trophic effects of GABA have been identified in the peripheral sympathetic nervous system. We have demonstrated that GABA and acetylcholine (ACh) may colocalize in the same axonal varicosities or be segregated into separate ones in the rat superior cervical ganglia (SCG). Neurotransmitter segregation varies with age and the presence of neurotrophic factors. Here, we explored age-dependent changes in the occurrence and segregation of GABA and ACh in rats ranging from 2 weeks old (wo) to 12 months old or older. Using immunohistochemistry, we characterized the expression of L-glutamic acid decarboxylase of 67 kDa (GAD67) and vesicular acetylcholine transporter (VAChT) in the rat SCG at 2, 4, 8, 12 wo and 12 months old or older. Our findings revealed that GAD67 was greater at 2 wo compared with the other ages, whereas VAChT levels were greater at 4 wo than at 12 wo and 12 months old or older. The segregation of these neurotransmitters was more pronounced at 2 and 4 wo. We observed a caudo-rostral gradient of segregation degree at 8 and 12 wo. Data point out that the occurrence and segregation of GABA and ACh exhibit developmental adaptative changes throughout the lifetime of rats. We hypothesize that during the early postnatal period, the increase in GABA and GABA-ACh segregation promotes the release of GABA alone which might play a role in trophic actions.

## 1. Introduction

Gamma-aminobutyric acid (GABA) functions as the primary inhibitory neurotransmitter in both the central and peripheral nervous systems. Its inhibitory effect is mediated through the binding of GABA to post-synaptic ionotropic GABA receptors, which, depending on the chloride electrochemical gradient, can lead to neuronal hyperpolarization and the suppression of action potential firing or membrane depolarization [1,2,3].

The presence of GABA has been identified in sympathetic preganglionic neurons (SPN). Wolff and colleagues initially detected GABA-like immunoreactivity in varicosities of preganglionic fibers within the superior cervical ganglion (SCG) of rats [4,5,6]. Ito et al. and our research group subsequently confirmed these findings by observing GABA markers in axonal varicosities within the SCG and in the cell bodies of SPN located in the spinal cord [7,8,9]. In mammalian autonomic ganglia, GABA exerts an inhibitory influence on cholinergic basal transmission [8,10,11], as well as on ganglionic long-term potentiation (gLTP), a recognized form of synaptic plasticity [12,13]. Furthermore, akin to its function in central nuclei [14], GABA also plays a trophic role, promoting synaptogenesis during neurodevelopment and aging in sympathetic ganglia [15].

We have demonstrated that GABA consistently colocalizes with acetylcholine (ACh) in the cell bodies of SPN. In contrast, in the axonal varicosities within the SCG, these transmitters may or may not coincide [8]. This differential routing of GABA and ACh to distinct varicosities exemplifies a common neuronal property referred to as neurotransmitter segregation, which involves the separate storage and release of neurotransmitters from different nerve terminals. This phenomenon of neurotransmitter segregation is observable in both central and peripheral neurons and enables the independent action of each separately released transmitter [16,17,18,19,20].

Our research has shown that neurotransmitter segregation is an adaptable phenomenon that can undergo plastic changes. For example, the segregation of ACh and met-enkephalin varies depending on neurotrophin levels [21,22], and similarly, the segregation of ACh and GABA increases in cold-stressed animals [9]. Additionally, we have observed that GABA expression, and its segregation from Ach, exhibits uneven distributions throughout the SCG, displaying rostro-caudal and caudo-rostral gradients, respectively, which correlate to the strength of transmission and synaptic plasticity observed in these ganglionic regions [8]. 

In our investigation of the sympathetic hyperactivity associated with hypertension, we observed that the presence of GABA and its segregation from ACh vary with age in both control normotensive Wistar Kyoto (WKy) rats and spontaneously hypertensive rats (SHR). We found a greater presence of GABA and increased segregation in young 6-week-old (wo) rats compared to adult rats at 12 wo [9]. Building upon these findings, in this work, we investigate using immunohistochemistry a potential age-related variance in the distribution and segregation of GABA and ACh across a wider age spectrum, encompassing neonatal rats to elderly rats aged 12 months or older.

## 2. Results

### 2.1. GAD67, VAChT, and Segregation Occurred in Preganglionic Varicosities

Consistent with our previous studies [8,9], we observed immunoreactivity (IR) to GAD67 and VAChT in the axonal varicosities of SPN located in SCG of Wistar rats across all age groups. GAD67-labeled varicosities appeared as scarce patchy markings either surrounding the cell bodies of principal ganglionic neurons or running between them, covering only 0.08 to 0.53% of the ganglionic area (Figure 1A–E,K). Conversely, VAChT-IR varicosities were distributed throughout the entire ganglion, covering an area between 2.30 to 3.62%, with most of them encircling the soma of ganglionic neurons (Figure 1F–J,L). Seven days after the decentralization of the ganglia (achieved by fully transecting the ipsilateral cervical sympathetic trunk), GAD67 and VAChT-IR were no longer detected, indicating their preganglionic origin.

We identified the segregation of VAChT and GAD67 in some preganglionic varicosities of the SCG in Wistar rats of all ages. Among all the GAD67-positive preganglionic varicosities, approximately 30% to 60% co-expressed VAChT, while 40% to 70% lacked VAChT labeling throughout their lifetime (Figure 2).

### 2.2. Expression of GAD67 and VAChT Depends on Age

We observed an age-dependent change in the number of positive varicosities for GAD67 or VAChT, with higher values typically found at early ages and lower values in adult and older rats (Figure 1). Specifically, the GAD67 content was significantly higher at 2 wo compared to all other ages. At this age, GAD67-positive preganglionic varicosities occupied circa 0.53% of the total ganglionic area, whereas at the other ages, the area occupied by GAD67-IR ranged from 0.20% at 4 wo to 0.03% at 12 months or older (Figure 1A–E,K). Similarly, VAChT content was significantly higher at 4 wo, reaching a ganglionic area of 3.62%, compared to areas of 2.30% and 2.17% at 12 wo and 12 months or older (Figure 1L).

### 2.3. Age-Dependent Variations in Segregation of GAD67 and VAChT

In line with our previous findings on SHR and WKy rats, we observed differences in the percentage of segregation of GAD67 and VAChT among rats of different ages. Greater segregation was noted at 2 and 4 wo, where 69.4% and 62.7% of the GAD67 positive varicosities lacked VAChT-IR, respectively (Figure 2A,B,F). The percentage of segregation decreased over the lifetime of rats; it was reduced to 48.1%, 43.0%, and 38.4% at 8 wo, 12 wo, and 12 months or older, respectively (Figure 2C–F). Additionally, at 4 wo, there was a significantly greater segregation than at 12 months or older (Figure 2F). The segregation exhibited a regional pattern with a caudo-rostral gradient in the ganglia of 8 and 12 wo rats (Figure 3).

## 3. Discussion

The presented data illustrate age-dependent changes in the expression and intracellular distribution of GABA and ACh within the axonal varicosities of SPN in the rat SCG. The expression of GAD67 and VAChT was highest during the early postnatal stages and gradually declined with age. The segregation of GAD67 and VAChT into different varicosities of individual neurons was also most prominent during early postnatal development (2 and 4 wo) and decreased from 8 wo onwards. Notably, despite variations in segregation, a caudo-rostral gradient distribution of segregation was consistently observed at 8 and 12 wo.

These age-dependent changes in GAD67 and VAChT expression, segregation, and the caudo-rostral gradient align with our prior findings of plastic changes in the SCG of SHR, WKy, and Wistar rats [8,9].

Age-dependent alterations in the presence and function of GABA have been observed in diverse neuronal structures. The GABAergic system undergoes critical developmental stages as it matures, including the transition from excitatory to inhibitory function and the emergence of spontaneous activity [23]. In the hippocampus of neonatal rats, granule cells express a GABAergic phenotype during the first three weeks of age, which later disappears, suggesting a developmentally regulated phenotype [24]. GABA excitatory action appears to play a role in appropriate behavior and cognitive function in adulthood, as blocking GABA-A receptors during the early developmental period (7–8 postnatal days; PND) leads to abnormalities in cognitive function, anxiety, pain processing, and neurobehavioral programming in adult life [25]. Wang et al. [26] also reported variations in GABA presence in newborn rats, with an increase in the number of GAD67-positive neurons in the CA1 hippocampal region from PND 13 to 24 compared to PND 11. An excellent review of age-dependent changes in GABA functions can be found in the work of Kilb [27]. However, developmental changes in the presence and function of GABA in sympathetic ganglia have not been previously reported. Recently, there has been a proposition that age can influence the overall function of the autonomic nervous system [28]. In addition to its synaptic modulatory effects, GABA trophic activity has been suggested to occur in sympathetic ganglia. It has been reported that exogenous prolonged GABA infusion induces the growth of spine-like expansions on dendrites and the production of free postsynaptic densities in the principal neurons of the rat SCG [15,29]. Consequently, it is possible that the increased presence of GABA in the SCG of 2 wo rats may play a role in the maturation of synaptic contacts.

The number of cholinergic varicosities also varies with age, although not as markedly as GABAergic ones. VAChT-IR varicosities are generally more abundant at earlier ages (4 wo) and decline in adulthood. These variations in the number of cholinergic varicosities may reflect a developmental and growth process, followed by age-related synaptic pruning. In the autonomic innervation of the gastrointestinal tract, an age-related loss of ACh-positive neurons starts in adulthood and progresses over the rodent’s lifetime [30]. Similar age-related decreases in cholinergic transmission have been reported, including impaired synthesis and the release of ACh in mouse brain slices [31] and a decrease in synaptic transmission at the neuromuscular junction of the phrenic nerve–diaphragm muscle in rats [32].

Concerning age-related differences in segregation degree, we previously observed that young SHR and WKy rats (6 wo) displayed higher levels of GABA-ACh segregation compared to adult rats (12 wo) [13]. Here, we confirm age-related differences in GABA-ACh segregation, with the highest degree of segregation observed in the early postnatal ages of 2 and 4 wo, followed by a decline as rats age. These age-dependent changes in segregation resemble the alterations induced by neurotrophic factors, such as ciliary neurotrophic factor (CNTF). In ganglionic neurons cocultured with cardiac myocytes, segregation between vesicular monoamine transporter (VMAT) and neuropeptide Y (NPY), as well as between VAChT and NPY, varies depending on the presence of CNTF [21]. Moreover, an endogenous increase in NGF, induced by the axotomy of ganglionic neurons in situ, leads to increased segregation of met-enkephalin and VAChT in preganglionic varicosities within the SCG of intact rats [22]. Considering these modulatory effects of neurotrophic factors on segregation, it is likely that a greater presence of neurotrophic factors at 2 and 4 wo could induce the increased segregation of ACh and GABA. Accordingly, age-dependent changes in segregation might be linked to the postnatal development of sympathetic ganglia, with greater segregation of ACh and GABA favoring independent GABA release, which may exert more trophic than synaptic actions during early postnatal stages. Notably, at 4 wo, a higher level of segregation remains, although GABA occurrence decreases compared to 2 wo, suggesting a slight reduction in trophic demand at this stage. The decline in GABA-ACh segregation may be associated with both a reduced demand for GABA trophic actions and the necessity for GABA modulation of cholinergic transmission across the lifetime of rats. In line with these notions, we have proposed an inverse correlation between GABA-ACh segregation and the inhibitory synaptic effect of GABA. This suggests that GABA, when colocalized and co-released with ACh from the same varicosities, could more effectively inhibit cholinergic transmission compared to situations where GABA is released separately from ACh [8].

Regarding the regional distribution of GABA expression and segregation, we have previously described a rostro-caudal gradient for GABA expression and an opposite caudo-rostral gradient for GABA-ACh segregation [8,9]. In the present study, we observed that the regional distribution of GABA-ACh segregation retains a caudo-rostral gradient in ganglia of rats at 8 and 12 wo. Conversely, GABA and ACh expression did not display regionalization under the conditions explored here.

## 4. Methods and Materials

### 4.1. Animals

Experiments were conducted using male Wistar rats of various ages: 2, 4, 8, and 12 wo, as well as females aged 12 months or older. In total 25 animals were used, 5 animals per age group. Animals were housed in the biological model unit of our institute and provided ad libitum feeding with a 12/12 light/dark cycle. According to Kilb [27], rats at 2 and 4 wo roughly correspond to human birth, 8 wo to the end of puberty, and 12 wo to adulthood beyond 20 years old [27] (see their Figure 2). All animal procedures were carried out in accordance with the ethical guidelines for the care and use of laboratory animals established by the National Academy of Sciences of the United States and were approved by our Institutional Committee for the Care and Use of Laboratory Animals.

### 4.2. Immunohistochemistry

For immunohistochemistry studies, rats were deeply anesthetized with sodium pentobarbital (125 mg/kg i.p.) and transcardially perfused with ice-cold phosphate buffer solution (0.01 M; PBS, pH 7.4), followed by ice-cold fixative solution (4% paraformaldehyde, 0.1 M PBS, pH 7.4). Both SCGs were dissected, unsheathed, post-fixed with paraformaldehyde, and cryoprotected in a 30% sucrose solution. Longitudinal sections of SCG were cut at a thickness of 12 µm using a cryostat (LEICA CM1520) at −20 °C and placed on gelatin-coated Superfrost slides (Electron Microscopy Sciences, Hatfield, PA, USA). Ganglia were sectioned along the z–axis obtaining 40 to 45 sections, and 9 of these sections were randomly selected for immunostaining. The sections were washed with 0.1 M PBS for 10 min, permeabilized, and blocked with a solution of 0.1 M PBS containing 0.3% Triton X-100 (PBS-Tx) and 10% normal donkey serum for 3 h at room temperature.

Tissue sections were incubated with the following primary antibodies (diluted in blocking solution, 5% normal donkey serum, 5% BSA, 0.3% Tx-100) for 16 h in a humid atmosphere at room temperature: goat polyclonal anti-vesicular acetylcholine transporter (VAChT; Immunostar, Hudson, WI, USA, Cat # 24286, 1:200 dilution) and mouse monoclonal anti-L-glutamic acid decarboxylase (GAD67, the enzyme responsible for the synthesis of GABA; Millipore-Sigma, Burlington, MA, USA, Cat # MAB5406, 1:200 dilution). Tissue sections were washed twice for 15 min each in PBS-Tx and then sequentially incubated with secondary antibodies for 2 h: donkey anti-mouse IgG CY3 (Jackson ImmunoResearch Lab Inc., West Grove, PA, USA, Cat # 715-165-151, 1:500 dilution), followed by donkey anti-goat IgG Alexa 488 (Jackson ImmunoResearch Lab Cat # 715-585-150, 1:200 dilution). After each secondary antibody incubation, the tissue sections were washed twice for 15 min each in PBS-Tx-100 and mounted on slides with fluorescence mounting medium (Dako, Santa Clara, CA, USA). Finally, the sections were examined with an epifluorescence microscope (Nikon Eclipse E600) equipped with appropriate filters for CY3 and Alexa 488. For each ganglion, one of the best-immunolabeled sections was randomly selected for confocal examination.

### 4.3. Image Acquisition and Analysis

Images of selected sections were acquired using a Nikon A1R+ laser scanning confocal head coupled to an Eclipse Ti-E inverted microscope (Nikon Corporation, Tokyo, Japan) equipped with a motorized stage (TI-S-E, Nikon) and controlled through Nis Elements C v.5.00 software. Tissue sections were analyzed under a PlanApo lambda 20X (N.A. 0.75) objective. Single-plane images were sequentially captured using standard galvanometric scanners and excitation wavelengths of 488 and 561 nm with AOTF modulation. We scanned and captured 40 to 50 images to cover the entire surface of each ganglion section.

To identify specific labels, in all the images acquired from each section examined, we selected the puncta optical density (OD) that exceeded the background level (i.e., OD > background mean + 2 SD) using the Metamorph image analysis system (v. 7.5.6; Universal Imaging Corporation, Molecular Devices, Downingtown, PA, USA). We then generated a mask (template) with the detected specific labeled varicosities. We quantified the number of pixels for each marker in labeled varicosities throughout the entire ganglion section and its distinct rostral and caudal regions. The area occupied by either VAChT or GAD67 was expressed as a percentage of the total section area. The degree of colocalization between VAChT and GAD67 labels was assessed by calculating the ratio of varicosities co-expressing both labels to the total number of GAD67-immunoreactive varicosities. The remaining varicosities, which expressed only GAD67, represented the percentage of segregation.

### 4.4. Statistics

The results from five independent rats for the age groups were expressed as mean ± SEM. Comparisons between regions in each age group were evaluated with an unpaired Student’s *t*-test. Multiple comparisons between all pairs of groups of different ages were assessed using a one-way ANOVA test followed by a Tukey post hoc test. *p*-values < 0.05 were considered statistically significant.

## 5. Conclusions

Our findings demonstrate the presence of the classic sympathetic transmitters GABA and ACh in preganglionic varicosities, either colocalizing within the same varicosity or segregated into different varicosities of single neurons, in the rat SCG from 2 wo onwards until old age. Segregation and the occurrence of GABA and ACh exhibit age-dependent changes, with higher levels generally observed at 2 and 4 wo compared to other ages. These findings shed light on the dynamic nature of neurotransmitter systems within the sympathetic nervous system and suggest potential roles for GABA and the segregation of GABA and ACh in different stages of development and adulthood. An increase in segregation may be linked to the greater presence of neurotrophic factors during early postnatal development. GABA detected at 2 wo may primarily exert trophic rather than synaptic activity. In conjunction with the elevated GABA levels at 2 wo, increased segregation at 2 and 4 wo promote the release of GABA without ACh, resulting in the greater availability of GABA with trophic effects that are highly demanded during this early postnatal stage. Further investigations are needed to uncover the functional significance and mechanisms underlying these age-dependent changes in sympathetic neurons.

## Figures and Tables

**Figure 1 ijms-25-02588-f001:**
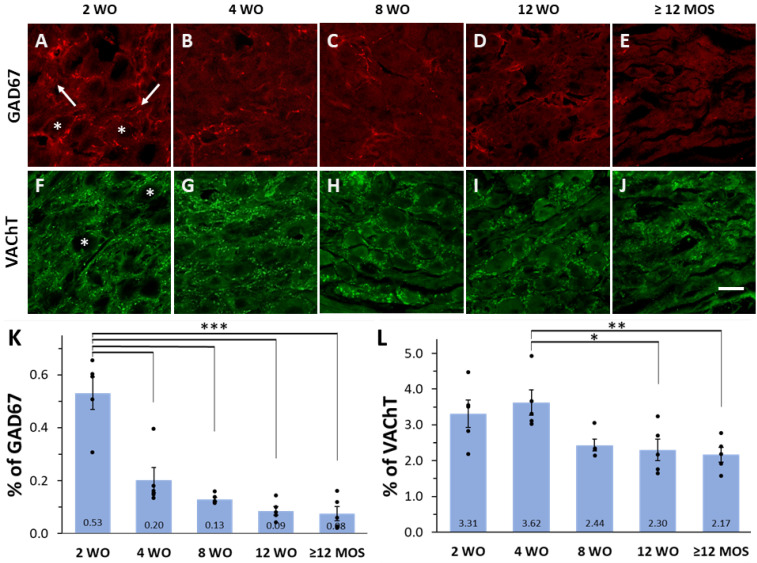
Expression of GAD67 and VAChT varies with rat age. (**A**–**J**) Micrographs of superior cervical ganglion (SCG) sections immunolabeled for GAD67 and VAChT at 2, 4, 8, and 12 wo, and 12 months old or older (≥12 MOS). As illustrated in panels (**A**,**F**), GAD67-IR varicosities were detected surrounding cell bodies (asterisk) or running between them (arrows), while VAChT labels were always detected surrounding cell bodies (asterisk). (**K**,**L**) Bar graphs of mean, standard errors, and the 5 individual values indicating the percent of ganglionic area occupied by the varicosities positive to GAD67 and VAChT at different ages. GAD67-IR was significantly greater at 2 wo (0.53 ± 0.06%) compared to 4 wo (0.20 ± 0.05%), 8 wo (0.13 ± 0.01%), 12 wo (0.09 ± 0.02%) and ≥12 MOS rats (0.08 ± 0.03%) (*** *p* < 0.001 for all comparisons). VAChT-IR was significantly greater at 4 wo (3.62 ± 0.35%) compared to 12 wo (2.30 ± 0.30%) and ≥12 MOS (2.17 ± 0.21%) (* *p* < 0.04 and ** *p* < 0.01, respectively). Calibration bar: 20 µm.

**Figure 2 ijms-25-02588-f002:**
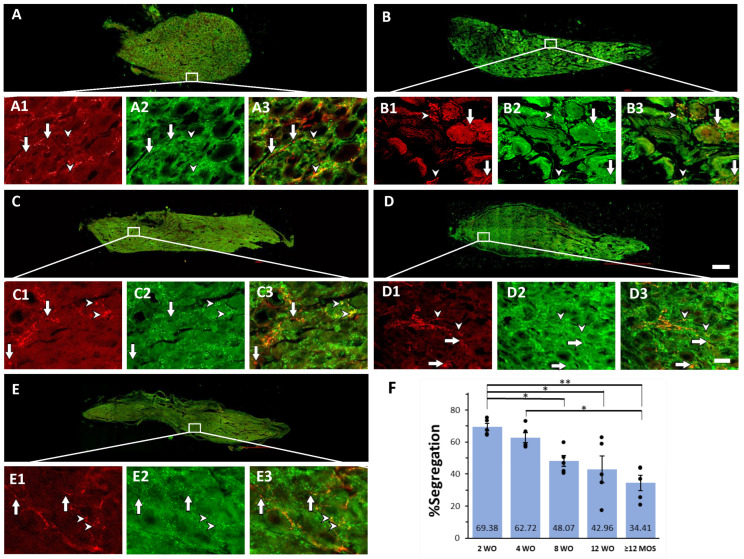
GABA-ACh segregation varies throughout the lifetime of rats, with greater segregation observed at neonatal ages of 2 and 4 wo. Micrographs of SCG sections double immunostained for GAD67 and VAChT at 2, 4, 8, and 12 wo, and ≥12 MOS. Panels (**A**–**E**) show the entire ganglionic areas of each age. The regions delimited by squares in each entire ganglia are enlarged in the corresponding panels, (**A1**–**A3**)–(**E1**–**E3**). These enlarged images enable us to note varicosities immunopositive to VAChT (1), to GAD67 (2), and merge images (3), either colocalizing (arrowheads) or segregated (arrows). Increased segregation was observed at 2 and 4 wo compared to the other ages. (**F**) Bar graph depicting the mean, standard errors, and the 5 individual values illustrating the degrees of GABA-VAChT segregation at the different ages. Segregation at 2 wo (69.38 ± 2.22%) was significantly greater than at 8 wo (48.07 ± 3.47%) and 12 wo (42.96 ± 8.27%) (* *p* < 0.04) and at ≥12 MOS (38.38 ± 4.57%) (** *p* < 0.01). Also, segregation at 4 wo, 62.72 ± 3.18% was larger than ≥12 MOS (* *p* < 0.04). Calibration bar: 20 µm for enlarged images and 200 µm for lower amplification images of complete ganglionic areas.

**Figure 3 ijms-25-02588-f003:**
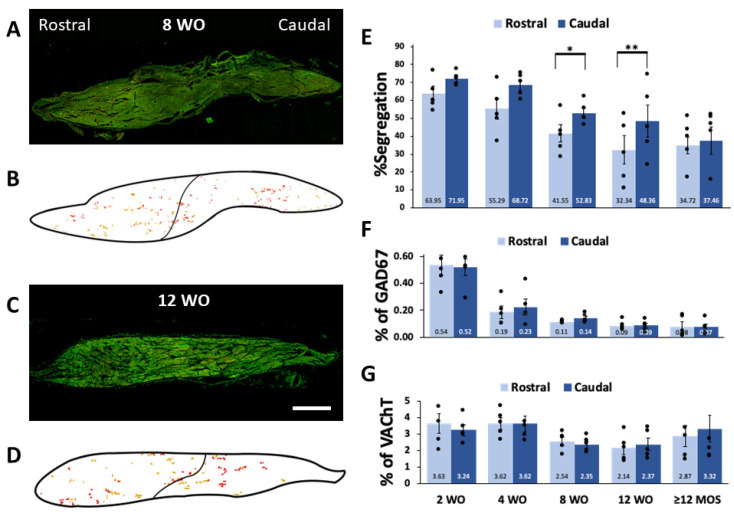
Segregation exhibits a heterogenous regional distribution, with a caudo-rostral gradient at 8 and 12 wo. To clearly illustrate differences in regional distribution of segregation in ganglia of 8 (**A**) and 12 wo (**C**) we generated schemes of the immunolabeled varicosities detected in pictures of entire ganglia shown in (**B**,**D**). Clusters of several varicosities immunolabeled to GAD67 colocalized or segregated from VAChT are represented as yellow and red dots, respectively. Schemes also show the border of ganglia (thick line) and the division between rostral and caudal regions (thin lines) Notice the greater presence of red dots representing varicosities of GAD67-IR lacking VAChT-IR in the caudal regions compared to the rostral regions. Bar graphs present mean, standard errors, and individual values of the regional distribution of GAD67-VAChT segregation (**E**) and of the occurrence of GAD67 (**F**) and VAChT (**G**) at 2, 4, 8 and 12 wo, and ≥12 MOS. Significant segregation differences were observed between ganglionic regions of rats at 8 and 12 wo. Segregation was greater in caudal regions at both ages: 8 wo (41.55 ± 4.73% rostral vs 52.83 ± 2.75 caudal; * *p* < 0.04) and 12 wo (32.34 ± 7.93% rostral vs. 48.36 ± 8.94% caudal; ** *p* < 0.01). The distribution of GAD67 and VAChT occurrence did not show significant regional differences. Calibration bar: 250 µm.

## Data Availability

The data presented in this study are available on request from the corresponding author.

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
