# Peer review of "Age-Dependent Changes in the Occurrence and Segregation of GABA and Acetylcholine in the Rat Superior Cervical Ganglia"

_ijms, 2024, doi:10.3390/ijms25052588_

Round 1
Reviewer 1 Report
Comments and Suggestions for Authors
Alfredo Hernandez and coworkers reported segregation of varicosities visualized with immunoreactivity to GAD67 and VAChT of sympathetic preganglionic neurons in the rat superior cervical ganglion. Their segregation the authors observed was changed in age-dependent manner. As they, in the manuscript, briefly show a relationship of axonal varicosities with rat ages, this reviewer believes this study contains new findings suitable for Brief Report in International Journal of Molecular Sciences. But I have several major comments about this manuscript as described below.
Major comments
1. The title and conclusion of this manuscript say “segregation of GABA and acetylcholine”; however, the authors analyzed immunoreactivity to GAD67. Is intracellular location of GAD67 the same as GABA? The reviewer read the three papers, i.e., Ito et al., 2007; Elinos et al., 2016; Merino-Jimenez et al., 2018 in the reference No. 7, 8 and 9, respectively. But these papers do not confirm colocalization of GABA and GAD67 by double immunostaining. Furthermore, GAD65, not GAD67, is localized on synaptic vesicles containing GABA, which is more reasonable to detect GABA localization. Is GAD67 is a proper marker to test GABA location? Thus, this reviewer is wondering about whether you surely examined segregation of GABA and acetylcholine.
2. This manuscript does not clearly show sample number in image analysis in the all figures, although the authors describe “5 individual values” in the figure legends. How many rats did you use? How many tissue sections from each rat? In sum, how many fluorescent images were acquired for image analysis? Please correctly describe information about them in Materials and methods.
3. In the section of Results, description of 2.1–2.3 and figure legends are overlapped. For example, the data values are shown in both the main text and figure legends. The authors need to revise them distinctly.
4. In the figure 2, the authors need to show individual images of red and green as well as their merged. In addition, you also describe what red and green images indicate, respectively.
5. It is difficult for this reviewer (probably as well as other readers) to understand the figure 3. (i) It hard to imagen what panel A and B indicate? What are the outer, thicker lines and middle, thinner lines? The authors may show raw images indicating these illustrations, (ii) you do not explain what the yellow dots are, and (iii) in addition to segregation (panel C), % of GAD67 and VAChT in rostral and caudal regions needs to be shown.
Author Response
R

Reviewer 2 Report
Comments and Suggestions for Authors
I find this is a well-structured study that presents interesting findings regarding segregation of neurotransmitters in the autonomic nervous system. The study is clearly presented allowing other scientist to attempt reproducing all findings.
However, I find an improvement could be made to the figures to provide readers with a wider context. That is, figures could incorporate lower magnification images of fluorescent samples, or HE stained sections to locate the anatomical structures that were stained and analyzed. This would be particularly helpful in figure 3, where only a representative scheme is provided.
Finally, I disagree with the conclusion where the authors link segregation with synaptic plasticity. I believe this is purely a developmental process, as supported by the evidence provided.
Author Response
R

Round 2
Reviewer 1 Report
Comments and Suggestions for Authors
The authors revised their manuscript in accordance with what I pointed out. This reviewer consider the manuscript is acceptable.